# 5β-Dihydrosteroids: Formation and Properties

**DOI:** 10.3390/ijms25168857

**Published:** 2024-08-14

**Authors:** Trevor M. Penning, Douglas F. Covey

**Affiliations:** 1Center of Excellence in Environmental Toxicology, Department of Systems Pharmacology & Translational Therapeutics, Perelman School of Medicine, University of Pennsylvania, Philadelphia, PA 19061, USA; 2Department of Developmental Biology, Washington University in St. Louis School of Medicine, St. Louis, MO 63110, USA; dcovey@wustl.edu; 3Taylor Family Institute for Innovative Psychiatric Research, Washington University in St. Louis School of Medicine, St. Louis, MO 63110, USA

**Keywords:** bile acids, farnesoid X receptor, neuroactive steroids, pregnane X receptor, smooth muscle relaxation, tocolysis

## Abstract

5β-Dihydrosteroids are produced by the reduction of Δ^4^-3-ketosteroids catalyzed by steroid 5β-reductase (AKR1D1). By analogy with steroid 5α-reductase, genetic deficiency exists in *AKR1D1* which leads to errors in newborn metabolism and in this case to bile acid deficiency. Also, like the 5α-dihydrosteroids (e.g., 5α-dihydrotestosterone), the 5β-dihydrosteroids produced by AKR1D1 are not inactive but regulate ligand access to nuclear receptors, can act as ligands for nuclear and membrane-bound receptors, and regulate ion-channel opening. For example, 5β-reduction of cortisol and cortisone yields the corresponding 5β-dihydroglucocorticoids which are inactive on the glucocorticoid receptor (GR) and provides an additional mechanism of pre-receptor regulation of ligands for the GR in liver cells. By contrast, 5β-pregnanes can act as neuroactive steroids at the GABA_A_ and NMDA receptors and at low-voltage-activated calcium channels, act as tocolytic agents, have analgesic activity and act as ligands for PXR, while bile acids act as ligands for FXR and thereby control cholesterol homeostasis. The 5β-androstanes also have potent vasodilatory properties and work through blockade of Ca^2+^ channels. Thus, a preference for 5β-dihydrosteroids to work at the membrane level exists via a variety of mechanisms. This article reviews the field and identifies gaps in knowledge to be addressed in future research.

## 1. Introduction

5β-Dihydrosteroids are products of the reduction of Δ^4^-3-ketosteroids catalyzed by steroid 5β-reductase (represented in humans by aldo-keto reductase 1D1 (AKR1D1)). The Δ^4^-3-ketosteroid functionality is found in all steroid hormones except the estrogens, meaning that 5β-dihydroglucocorticoids, 5β-dihydropregnanes, and 5β-dihydroandrostanes exist. In addition, the Δ^4^-3-ketosteroid is present in cholestenes (e.g., 7α-hydroxy-cholesten-3-one) and are important intermediates in bile acid biosynthesis. 5β-Dihydrosteroids have unique structural properties in that the A/B rings of the steroid are now in the *cis*-configuration which converts the planar steroid structure to one that has a 90° bend, Figure 1.

Δ^4^-3-ketosteroids are metabolized by steroid 5α-reductases (SRD5A1 and SRD5A2) to give rise to 5α-dihydrosteroids [1], and by steroid 5β-reductase [2] to give rise to 5β-dihydrosteroids, thus as a result of double bond reduction, two isomers arise. The corresponding 5α- and 5β-dihydrosteroids all share a 3-ketone group which can then be further reduced by 3α- or 3β-hydroxysteroid dehydrogenases to give rise to four stereoisomeric tetrahydrosteroids: 3α,5α-, 3β,5α-, 3α,5β- and 3β,5β-tetrahydrosteroids. The 3α- or 3β-hydroxysteroid dehydrogenases responsible for 3-ketosteroid reduction are also members of the AKR superfamily, and these reactions are catalyzed by AKR1C1 and AKR1C2, respectively. In the case of liver metabolism, AKR1C4 is also prominent in the formation of the 3α,5β-tetrahydrosteroids [3,4]. This means that each Δ^4^-3-ketosteroid can give rise to six metabolites, Figure 2.

A large portion of the literature has focused on the formation of 5α-dihydrotestosterone in peripheral tissues since it has a higher potency for the androgen receptor than testosterone [1]. In addition, *SRD5A2* deficiency and SNPs have documented the importance of 5α-reduction in genetic disorders in newborns [6,7]. By contrast, it was felt that 5β-dihydrosteroids and their tetrahydrosteroid metabolites were inert. However, steroid 5β-reductase plays a pivotal role in bile acid biosynthesis where bile acids are important for the absorption of lipids and fat-soluble vitamins, and genetic deficiency in the *AKR1D1* gene leads to bile acid deficiency [8,9,10]. Bile acids also act as ligands for FXR [11,12], which controls cholesterol homeostasis as the major source of bile acids and act as ligands for PXR which regulates the expression of P450 enzymes [13]. Thus, just like *SRD5A2* has a well-documented genetic deficiency and regulates ligand access to a nuclear receptor, the androgen receptor, *AKR1D1* has an associated genetic deficiency and regulates ligand access to the nuclear receptors, FXR and PXR.

The last review on AKR1D1 and 5β-dihydrosteroids was published by one of our laboratories in 2014 and included an account of the cardiac glycosides and ecdysone [14]. This review is focused on mammalian steroid 5β-reductases required to synthesize 5β-dihydrosteroids as well as the biological functions of these steroids and their downstream metabolites. 

## 2. Steroid 5β-Reductase Gene

The human gene for *AKR1D1* is located on chromosome 7. It consists of nine exons which can give rise to four splice variants: AKR1D1-0002 (encodes for a protein of 326 amino acids and is wild type or full length AKR1D1); AKR1D1-006 (encodes for a protein of 290 amino acids and lacks exon 8); AKR1D1-0001 (encodes for a protein of 285 amino acids and lacks exon 5), and AKR1D-009 (encodes for a 96-amino acid truncated protein) (Figure 3) [15]. Although these truncated proteins can be expressed in HEK-293 cells, they are unable to metabolize cortisol or prednisolone [15]. The stability of these proteins could be increased with a proteasome inhibitor MG-132 suggesting that they are targeted for proteasomal degradation. The transcript for AKR1D1-0002 (wild type) is predominantly expressed in the liver [15].

## 3. Steroid 5β-Reductase Enzymology

Rat Δ^4^-3-ketosteroid 5β-reductase was first purified to homogeneity from liver as a 37-kDa protein [16]. The cDNA for the corresponding human enzyme was subsequently cloned, but initially characterized only in mammalian cell expression systems [17]. Sequencing of the cDNA showed that the enzyme was a member of the aldo-keto reductase (AKR) superfamily of proteins and because of its low sequence identity (<60% with other members) was assigned to its own sub-family and named AKR1D1 [18]. Rat and murine steroid 5β-reductase correspond to AKR1D2 and AKR1D4, respectively. Unlike other AKRs, AKR1D1 does not catalyze carbonyl reduction but instead catalyzes steroid double bond reduction expanding the substrate specificity of this protein superfamily. Expression in *E.coli* led to milligram amounts of the protein which provided sufficient protein to characterize its substrate specificity which showed that a single enzyme could produce C19, C21, C24 and C27 5β-dihydrosteroids [2]. A detailed purification scheme for AKR1D1 has been recently published by one of our laboratories [19]. The abundance of the protein also permitted the elucidation of X-ray crystal structures of ternary complexes of AKR1D1.NADP^+^.testosterone, AKR1D1.NADP^+^. progesterone, and AKR1D1.NADP^+^_._cortisone [20]. The latter two complexes showed productive binding modes for the steroid and provided an explanation as to why the enzyme catalyzed double bond reduction rather than carbonyl reduction. In these structures, the catalytic H120 was replaced with E120. This substitution permitted the steroid to bind deeper into the steroid pocket so that C5 was in proximity to NADPH so that it can receive a hydride ion from the C4 position of the cofactor. In addition, the side chain of E120 was found to be in an *anti*-conformation and fully protonated, permitting it to work as a superacid to promote enolization of the Δ^4^-3-ketosteroid and facilitate hydride transfer [20,21]. The importance of 5β-dihydrosteroids in physiology is revealed in the phenotype of 5β-reductase deficiency and the phenotype of *AKR1D4* knockout mice.

## 4. Steroid 5β-Reductase Deficiency

Inherited *AKR1D1* deficiency is an autosomal recessive disorder, which is now recognized as congenital bile acid synthesis defect type 2, CBAS 2 (OMIM 235555) [8,9,10]. AKR1D1 deficiency is fatal to neonates unless diagnosed early and underlines the importance of proper bile acid biosynthesis [22]. Bile acids have the A/B *cis*-ring configuration to generate an amphipathic structure in which the β-face is non-polar and the α-face is polar, giving rise to their emulsifying properties. These properties are required for the absorption of lipids and fat-soluble vitamins (A, D, E and K). In addition, the absence of bile acids prevents the negative feed-back inhibition of the rate-limiting step in bile acid synthesis catalyzed by cholesterol 7α-hydroxylase [23]. This leads to the accumulation of C_27_ bile acid precursors bearing intact Δ^4^-3-oxo groups, which, upon reduction by steroid 5α-reductase, are converted to *allo*-bile acids [10]. This situation is also exacerbated by the depletion of ligands for FXR which will result in an increase in cholesterol biosynthesis. The *allo*-bile acids are hepatotoxic, cause cholestasis and their accumulation results in liver failure. Steroid 5β-reductase deficiency is characterized by seven single-point mutations in the enzyme (L106F, P133R, P198L, G223E, D241V, R261C, and R266Q) as well as one frame shift mutation and two nonsense mutations [9,10,24,25]. The seven point mutations occur in evolutionary conserved positions on the (α/β)_8_ barrel in the AKR superfamily, suggesting that they play roles in stabilizing the protein fold. When these amino acid substitutions are mapped to the available crystal structures of AKR1D1, they are not found at the active site, the NADPH binding site, or the steroid binding site. Instead, these mutations appear to affect enzyme stability, as judged by the difficulty in expressing their corresponding cDNAs in *E. coli* and HEK-293 cells [26]. The exception was the P133R mutant which could be purified to homogeneity and was shown to cause a 40-fold increase in *K*_d_ values for the NADP(H) cofactors [27]. 

The defect in bile acid biosynthesis can be corrected by the oral supplementation of bile acids in the diet of affected individuals [28]. However, in one case the bile acid supplementation could be discontinued, suggesting an adaptative response [29]. Whereas it is known that the microbiota play an important role in the synthesis of secondary bile acids (deoxycholic acid and lithocholic acid), it is unknown whether the microbiota were able to compensate for the defect in primary bile acid synthesis or whether they express their own steroid 5β-reductase. 

## 5. AKR1D4 Knockout Mice

The murine homolog of 5β-reductase is known as AKR1D4, and the importance of its physiological function may be inferred by its genetic knockdown [30]. Mature (30 week) male and female *AKR1D4*^−/−^ mice had decreased total hepatic and serum bile acids as expected. *AKR1D4*^−/−^ mice showed a sexually dimorphic effect on hepatic bile-acid metabolizing genes that was reflected in the levels of bile acid intermediates. When male *AKR1D4*^−/−^ mice were challenged with a high-fat diet, they were more insulin tolerant and liver and adipose tissue had less lipid accumulation but had increases in serum triglyceride and increased intramuscular triacylglycerol [30]. 12α-Hydroxylase (*Cyp8b1*) expression increased in females but not in males and was accompanied by increases in the AKR1D4 substrates 7α,12α-dihydroxy-4-cholest-3-one and 7α-hydroxy-4-cholest-3-one. A reduction in 12α-hydroxylated bile acids as noted in male mice may provide an explanation for their improved insulin tolerance. 

## 6. Glucocorticoids and 5β-dihydroglucocorticoids

AKR1D1 can play an important role in glucocorticoid clearance by producing 5β-dihydroglucocorticoids, suggesting that it may control ligand access to the glucocorticoid receptor (GR) [31]. The conversion of cortisone (inactive hormone) to cortisol (active hormone) is governed by 11β-hydroxysteroid dehydrogenase type 1 (*11HSDB1*) while the reverse reaction is catalyzed by 11β-hydroxysteroid dehydrogenase type 2 (*11HSDB2*) [32]. These enzymes are regarded as the molecular switches that are involved in the pre-receptor regulation of ligands for GR. However, both cortisol and cortisone can be converted to 5β-dihydrocortisol and 5β-dihydrocortisone which provides another mechanism of ligand control, Figure 4 [31]. AKR1D1 can regulate GR action in both HepG2 cells and HEK-293 cells in reporter gene assays. However, the effect in HepG2 cells was quite modest. By contrast, si-RNA for AKR1D1 reduced cortisone clearance in HepG2 cells, substantially supporting its role in the pre-receptor regulation of GR in liver cells.

Expression of the *AKR1D1* gene is also repressed by dexamethasone and can drive changes in gluconeogenesis and glycogen synthesis [33]. The effects of dexamethasone on *AKR1D1* gene expression are mirrored by genetic knockdown of *AKR1D1.* These findings suggest that downregulation of *AKR1D1* would increase hepatic glucose output and exacerbate type 2 diabetes [33]. The ability of synthetic glucocorticoids to repress *AKR1D1* expression raises the possibility that steroidal anti-inflammatory drugs may potentiate their pharmacological effects by inhibiting their own metabolism. This concept was examined in rat liver extracts almost 40 years ago where, even in the presence of indomethacin which inhibited 3α-hydroxysteroid dehydrogenase by over 90%, 5β-reduced glucocorticoids were still rapidly metabolized to the tetrahydrosteroids [34]. These kinetic measurements indicate that even in the presence of low steroid 5β-reductase, the flux favors the formation of the tetrahydrosteroids.

Unexpectedly, 5β-dihydrocortisol potentiates the action of dexamethasone to increase intraocular pressure in rabbits and in humans and may contribute to primary open angle glaucoma (POAG) [35,36]. Importantly, 3α,5β-tetrahydrocortisol was found to be a natural antagonist of this effect [37]. It is likely that this physiological antagonism is mediated by an increase in the relaxation of the trabecular meshwork to allow drainage of the aqueous humor. A small clinical trial showed that 3α,5β-tetrahydrocortisol was effective in lowering intraocular pressure in patients with POAG [37]. The hypotensive effect takes 3–7 days to occur. This was one of the earlier reports of membrane effects of 5β-dihydrosteroids and others followed.

## 7. Progestins and 5β-Pregnanes

### 7.1. Neuroactive Steroids

5β-Pregnanes derived from progesterone can act as neuroactive steroids (NAS) [38,39]. The term NAS is preferred over neurosteroids which would imply that 5β-pregnanes would be biosynthesized in the CNS via neurosteroidogenesis [40,41], when there is no evidence for the expression of AKR1D1 in human brain. The involvement of AKR1C enzymes in the subsequent formation of 5β-pregnanes, pregnanolone and pregnanediol, and their epimers has been elucidated using recombinant enzymes and product profiling by mass spectrometry and steady state kinetic measurements, Figure 5 [42]. Since progesterone can be converted to 20α-hydroxyprogesterone by AKR1C1, 3α,5β- and 3β,5β-tetrahydropregnanes are possible from 5β-dihydroprogesterone, and 3α,5β- and 3β,5β-tetrahydropregnanes are also possible from 20α-hydroxy-5β-pregnane-3,20-dione giving rise to four isomers. If this process is repeated for 20β-hydroxyprogesterone, another four isomers can be produced resulting in eight steroids in total. Although there is little evidence for a 20β-HSD in humans, there is a synthetic opportunity to make these steroids and determine their bioactivity. 

Structures of bioactive 5β-dihydrosteroids including pharmacological agents are shown in Figure 6. Depending on their structures, 5β-pregnanes can potentiate chloride conductance at GABA_A_ receptors and potentiate the inhibitory effects of GABA [43]. Whereas pregnenolone-SO_4_ (PES) is known to stimulate N-methyl-D-aspartate (NMDA) receptors and potentiate the excitatory effects of glutamate [44], the opposite is true for pregnanolone-SO_4_ (PAS) [45]. 5β-Pregnanes can also modulate the activity of the muscarinic acetylcholine receptors (mAcHR) [46].

Both the synaptic and extrasynaptic GABA_A_ receptors contain an allosteric steroid site that binds allopregnanolone (3α-hydroxy-5α-pregnane-20-one) to increase chloride conductance and increase anxiolytic activity [47]. The ability to work on both types of receptors may provide an advantage over the benzodiazepines that exert their activity only on certain types of synaptic GABA_A_ receptors. 

The GABA_A_ heteropentameric receptor consists of five subunits selected from 19 different subunits, but most often consists of two α-subunits, two β-subunits and one γ-subunit. One prominent isoform is composed of α1, β2, and γ2 subunits which are arranged γ2β2α1β2α1 counterclockwise around a central pore. NAS that are positive allosteric modulators bind to the α-subunit and at the β+/α- subunit interface whereas benzodiazepines bind to the γ/α-subunit interface [48]. 

Structure activity relationships demonstrate that the presence of the 3α-hydroxy group is preferred for GABA_A_ receptor activity [38]. However, both 5β-dihydroprogesterone and 3α,5β-tetrahydroprogesterone can modulate the GABA_A_ receptor as well. The resulting chloride currents seen with 5β-dihydroprogesterone were potentiated by diazepam and phenobarbital and attenuated by bicuculline, a GABA-receptor antagonist [43]. Structure activity relationships showed that 3α,5β-tetrahydroprogesterone (pregnanolone) had a less potentiating effect than allopregnanolone on flunitrazepam binding, Figure 6 [38]. In TBPS (*tert*-butylbicyclophosphorothionate) displacement assays, allopregnanolone and pregnanolone were essentially equipotent [49], and both bind to the same intersubunit binding site on the GABA_A_ receptor [50]. To exploit this finding, a formulation of pregnanolone (eltanolone) was evaluated as a general anesthetic but was not carried forward for clinical use [51]. These findings are consistent with the view that the configuration at C5 is less critical for GABA_A_ receptor activity than the presence of the 3α-hydroxyl group. SAGE pharmaceuticals has developed an active NAS program and their clinically approved drug SAGE-217 (zuranolone 3α-hydroxy-3β-methyl-21-(4-cyano-1H-pyrazol-1-yl)-19-nor-5β-pregnan-20-one) for the treatment of post-partum depression has a 5β-configuration [52]. By contrast, epipregnanolone (3β-hydroxy-5β-pregnan-20-one) can exert potent peripheral analgesia and blocks T-type (low-voltage activated) calcium channels while sparing the GABA_A_ receptor, showing that it is possible to separate out these two activities with isomers of 5β-pregnanes [53].

NAS not only potentiate the effects of the inhibitory neurotransmitter GABA but also modulate the effects of the excitatory neurotransmitter receptors: α-amino-3-hydroxy-5-methyl-4-isoxazoepeorpionic acid (AMPA) and N-methyl-D-aspartate (NMDA) glutamate receptors. NMDA receptors are heterotetramers and are composed of different combinations of GluN1, GluN2 (A-D) and GluN3 (A-B) subunits [54]. Pregnanolone and PAS potentiate the presynaptic release of glutamate [55]. Pregnanolone sulfate also reduces single channel opening from individual GluN1/GluN2 receptors, providing a mechanism for desensitization of NMDA receptors [56]. In the model proposed, resting GluN1/GluN2 receptors have no affinity for PAS; however, exposure to glutamate transitions the receptors to an active state that binds PAS which leads to reduced channel opening [56]. The ability of pregnanolone-3-conjugates to act as NAS has led to medicinal chemistry efforts to modulate the activity of NMDA receptors. This effort led to the synthesis of 4-(20-oxo-5β-pregnan-3β-yl)-butanoic acid (EPA-But) which could act with PAS in an additive manner suggesting they interact differently with NMDA receptors [57]. These findings are consistent with the requirement for the bent steroid structure in 5β-pregnanes for NMDA inhibition, whereas the more planar ring structure associated with pregnenolone and PES favors potentiation of NMDA-mediated Ca^2+^ increases and neuronal cell death. Hemiesters of various lengths can substitute for the sulfate group of the positive modulator PES and the negative regulator PAS [45].

5β-Dihydropregnanes can also bind to the muscarinic (M1, M2, M4) AcHR with affinities around 100 nM and are lower than that observed for their 5α-dihydro analogues. Pregnanolone binds to the M1, M2 and M4 receptors and epipregnanolone binds to the M4 receptor with affinities between 200–350 nM [46]. This property is also conserved with the 5β-dihydroandrostanes. Thus, steroidal MS-112 (17-methylene-5β-androstan-3α-yl-3-hemiglutarate) binds to the allosteric site with a *K*_i_ of 16 nM: a value obtained with the displacement of radiolabeled *N*-methyl-[^3^H] scopolamine in cell membranes from rat brain tissues [46].

The neuromodulatory activity of endogenous 5β-reduced pregnanes may depend on whether they are produced in different brain regions and would require the expression of AKR1D1 as well as the corresponding 3α-hydroxysteroid dehydrogenase (AKR1C2) to produce the 3α,5β-tetrahydrosteroid. In the case of pregnanolone sulfate, the corresponding sulfotransferase would also need to be expressed but the evidence that all these enzymes are expressed in the CNS to synthesize 5β-pregnanes is not strong [58]. However, these steroids would be present in the systemic circulation and could cross the blood-brain barrier to influence receptor activity. 

Information relating to the SAR for the 5β-configuration to exert NAS activity comes from detailed studies on synthetic *ent*-steroids [59]. *ent*-Steroids, unlike the diastereomers (allopregnanolone and pregnanolone) are mirror images of these natural steroids, they do not occur naturally and require total synthesis, Figure 7. Nevertheless, they are important structural probes to distinguish between enantioselective actions (often receptor-mediated) versus those that are due to identical physio-chemical properties shared by the natural steroid and the *ent*-steroid. Thus, potentiation of GABA_A_ and GABA_C_ receptors, and inhibition of T-type (low voltage-activated) calcium channels is enantioselective as might be expected from receptor binding. In some instances, these activities can be separated using synthetic NAS. For example, 3β-Hydroxy-5β-androstane-17β-carbonitrile does not directly modulate neuronal GABA_A_ receptors but blocks T-type calcium channels [39]. However, effects on lipid bilayers, plasma membrane accumulation, intracellular accumulation in the Golgi, neuroprotection, and pregnane X-receptor activation are all non-enantioselective, Figure 8. 

### 7.2. Tocolytic Hormones

5β-Dihydroprogesterone is also produced in the uterus. Steroid 5β-reductase mRNA is expressed in rat and human uterus, and 5β-dihydroprogesterone was found to cause rapid relaxation of uterine smooth muscle suggesting a role for this metabolite as a tocolytic hormone in human pregnancy. The authors concluded that 5β-dihydroprogesterone increased iNOS-modulated uterine tone [61]. However, others showed that 5β-dihydroprogesterone could inhibit oxytocin-mediated contraction when applied to uterine smooth muscle strips. 5β-Dihydroprogesterone was shown to bind directly to the oxytocin receptor in competitive binding assays and replace [^3^H]-oxytocin. Importantly, this effect was observed only with the human oxytocin receptor and not the rat receptor, see Table 1 [62]. It was also found that serum levels of 5β-dihydroprogesterone as measured by LC-MS fell in women undergoing the onset of spontaneous labor and this correlated with a decrease in AKR1D1 expression in human placenta [63,64]. Others showed that oxytocin modulates iNOS in human fetal membranes at term, suggesting that iNOS was ultimately responsible [65].

### 7.3. Platelet Activation

Another unexpected property of the 5β-pregnanes was their ability to stimulate Ca^2+^ influx into human platelets leading to their activation, see Table 2 [66]. Structure–activity relationships demonstrate that pregnanediol (5β-Pregnane-3α,20α-diol) was more efficacious than isopregnanediol (5α-Pregnane-3β,20α-diol) indicating that the configuration of the tetrahydrosteroid was important for maximal activity. One caveat to these experiments was the relatively high concentrations of steroids (10 μM) used to observe these effects, which questions their physiologic significance. 

## 8. 5β-Androstanes

5β-Dihydrotestosterone (5β-DHT) was found to produce substantial systemic hypotensive and antihypertensive responses in normotensive and hypertensive male rats and similar effects were seen in a model of preeclampsia [67]. These findings suggest that 5β-DHT may be an important regulator of blood pressure during pregnancy [65]. 5β-DHT may be derived from AKR1D1 expression in the placenta and from the fetal liver. 5β-DHT was a potent vasodilator in isolated aortas of hypertensive pregnant rats and isolated aortas of normotensive rats. Concentration response curves indicated that vasorelaxation of KCl-induced contraction was more sensitive to androgens in the 0.1–100 μM range than phenylephrine, suggesting blockade on L-VOCCs (low voltage-operated calcium currents) [67]. 

The relaxation of blood vessels observed with 5β-DHT is consistent with findings in human umbilical artery, rat aorta, canine coronary and femoral artery and the saphenous vein [65]. The ability of 5β-DHT to relax smooth muscle cells is not limited to blood vessels. 5β-DHT displays non-genomic rapid relaxation of carbachol or antigenic challenge in pre-contracted guinea pig airway smooth muscle [66], suggesting a potential role in the modulation of asthmatic symptoms. A proposed mechanism of smooth muscle relaxation involves blockade of L-type voltage-dependent Ca^2+^ channels, stored operated Ca^2+^ channels, IP3 receptors and promotion of PGE_2_ synthesis [68].

The 3α,5β-androgen etiocholanolone (3α-hydroxy-5β-androstane-17-one), also known as 5β-epiandrosterone, has been used as a biomarker in serum measurements to detect steroid 5α-reductase deficiency which is accompanied by a decrease in androsterone:5β-epiandrosterone ratios [69]. Knowing that indomethacin inhibits AKR1C3, increases in urinary androsterone:5β-epiandrosterone and 5α-Adiol:5β-Adiol ratios were noted in anti-doping laboratories in individuals on NSAIDs most likely due to enhanced reduction of 5α/5β-androstane-3,17-dione by AKR1C2 [70]. Etiocholanolone has biological properties of its own. It is known to induce steroid fever by activating the inflammasome [71]. In addition, it plays an indirect role in erythropoiesis mediated by androgens, where it induces heme biosynthesis in the liver through the induction of δ-aminolevulinate synthase [72,73]. Etiocholanolone-like pregnanolone can also regulate the activity of the GABA_A_ receptor and can act as an anticonvulsant for epileptic seizures [49,66]. The enantiomer (*ent*-steroid) of etiocholanolone has enhanced activity relative to etiocholanolone as a positive allosteric modulator of the GABA_A_ receptors and has been shown to have anticonvulsant activity in mice [49,74].

## 9. Farnesoid X Receptor (FXR) Ligands

5β-Dihydrosteroids can also act as ligands for the nuclear receptor FXR. Chenodeoxycholate (5β-cholanic acid 3α,7α-diol), cholic acid (5β-cholanic acid 3α,7α,12α-triol), deoxycholic acid (5β-cholanic acid 3α,12α-diol) and lithocholic acid (3α-hydroxy-5β-cholanic acid) activate FXR [11,12,75]. During their transit to the small intestine, chenodeoxycholate and cholic acid undergo 7α-dehydroxylation and are converted to the secondary bile acids lithocholic acid and deoxycholic acid, respectively. Structure activity relationships demonstrate that the configuration of the substituent at the 7-position is crucial so that bile acids show the following rank order: 7α-OH >> 7-keto >> 7β-hydroxy for the activation of FXR [76]. The ligand-activated FXR recruits a co-activator which in turn leads to the induction of a repressor of the transcriptional activity of the oxysterol receptor LXR which is a positive regulator of *CYP7a* (7α-hydroxylase), the rate-limiting step in bile acid biosynthesis. This provides a mechanism of positive feedback inhibition of bile-acid production [76,77]. Enantiomeric forms of lithocholic acid, chenodeoxycholate, and deoxycholate have been synthesized and their properties compared to the natural bile acids. Both *ent*-bile acids and natural bile acids gave similar critical micelle concentrations showing their physio-chemical properties to be similar. However, they had differential effects on bile-acid induced apoptosis in colon cells, where the natural bile acids were better inducers of apoptosis [78]. 

## 10. Pregnane X Receptor (PXR) Ligands

5β-Dihydrosteroids can act as ligands for the nuclear receptor PXR. PXR is a xenosensor that is in the nucleus, and when ligands bind, its corepressors dissociate to enable transcription of target genes in the liver [79]. One of the most upregulated genes is *CYP3A4* [79], which mediates the metabolism of 50% of all drugs. PXR also regulates the expression of other genes involved in xenobiotic metabolism, including *CYP3A4, CYP2B6, CYP2C9* [80,81], as well as genes critical to bile acid metabolism. The most potent ligands for PXR are C21 steroids (pregnanes), such as 5β-pregnane-3,20-dione, Table 3 [82]. 5β-Androstan-3α-ol is even more potent and is as efficacious as 5β-pregnane-3,20-dione but it is not an endogenous steroid. Of the bile acid ligands, the secondary bile acids lithocholic acid and lithocholic acetate are among the most potent and efficacious, raising the prospect that 5β-steroid metabolites produced in the microbiota can regulate drug metabolism in the liver. However, there are distinct differences in PXR ligand specificity depending on species. For example, 5β-cholestane-3α,7α,12α-triol is a potent PXR ligand only in mice [79], and it accumulates in *cyp27* deficiency in mice leading to overexpression of *cyp3a4*.

## 11. Discussion

5β-Dihydrosteroids are not inert metabolites and display a range of bioactivities. Often this activity is rapid and involves non-genomic signaling. Moreover, evidence that these effects are mediated by the now-accepted membrane-bound steroid hormone receptors is not strong, but in part may be related to the scarcity of studies to determine whether the 5β-dihydrosteroids can mediate effects through these receptors. 

NAS have the capacity to bind to allosteric sites on membrane-associated receptors that control channel opening and can have opposite effects depending on whether they bind to the GABA_A_ receptor to promote chloride conductance in the presence of GABA (e.g., allopregnanolone) [47], or stimulate the NMDA ionotrophic glutamate receptor (e.g., pregnenolone-SO_4_) to increase Ca^2+^ channel opening. 5β-Pregnanes can also act as allosteric regulators of GABA_A_ (e.g., pregnanolone) or in this instance inhibit the NMDA receptor (e.g., pregnanolone and pregnanolone-SO_4_). Evidence for the de novo synthesis of Δ^4^-3-ketosteroids to form NAS has been controversial. This may not occur in the human temporal lobe and limbic system due to the lack of cytochrome P45017c and 3β-HSD/isomerase expression, but reduction of 5β-dihydrosteroids to the corresponding tetrahydrosteroids may occur due to the local expression of either AKR1C enzymes or other 3-ketosteroid reductases [58]. Others have demonstrated the presence of steroidogenic enzymes at the mRNA, and protein level and in some cases measured activity in brain slices [40]. 

To synthesize neuroactive 5β-dihydrosteroids in the CNS the presence of AKR1D1 would be required. Thus far evidence supports AKR1D1 expression in the liver and placenta suggesting that the neuroactive 5β-pregnanes would have to originate from these sources and enter the systemic circulation and then cross the blood-brain barrier. The fact that 5β-pregnanes could originate from the placenta suggests there may be some gender differences in neuronal activity, not unlike changes seen in allopregnanolone production post-partum. Many of these uncertainties could be tackled by the precise and accurate measurement of 5α- and 5β-pregnane metabolites of progesterone in discrete brain regions and in the CSF, but this would require the development of LC-MS strategies to separate out the large number of regio- and stereoisomers that could potentially form.

In some instances, other membrane receptors can be engaged, for example 5β-dihydroprogesterone can bind to the oxytocin receptor and antagonize the effect of oxytocin to prevent uterine smooth muscle contraction. This property seems unique to 5β-dihydroprogesterone but the number of 5β-pregnane metabolites that mediate Ca^2+^ uptake in platelets is many. Thus, a gap in knowledge exists in not knowing how many 5β-pregnanes may have biological activity without a more systematic approach.

When these receptors are engaged, structure-activity studies suggest that stereochemistry of the steroid is important. 5β-Dihydrosteroids have a bent shape, and this physicochemical property may result in membrane perturbation and affect ion channel opening without receptor interaction. These non-receptor-mediated effects can be distinguished from the receptor-mediated effects using the elegant *ent*-steroid approach developed by the Covey laboratory [59]. The potential therapeutic actions of NAS including *ent*-steroids has been recently reviewed [60]. 

Apart from working at the membrane, 5β-dihydrosteroids can act as ligands for FXR and PXR which regulate cholesterol homeostasis and work as xenosensors to regulate drug metabolism, respectively.

Collectively, these studies suggest that 5β-dihydrosteroids and analogs of their downstream metabolites have therapeutic potential. The development of brexanolone (cyclodextrin formulation of allopregnanolone for post-partum depression) led to the development of zuranolone for use in major depressive disorder (SAGE Therapeutics). Unlike allopregnanolone, zuranolone is a 3α-hydroxysteroid with a 5β-configuration. Other synthetic opportunities exist to fully explore the properties of 5β-dihydrosteroids and their downstream metabolites. For example, non-metabolizable 5β-dihydroprogesterone analogs could be developed to maintain pregnancy and act as tocolytics. In addition, the synthesis of hemiesters of 3β,5β-tetrahydropregnanes with different aliphatic side chains could be further explored to modulate NMDA and GABA_A_ receptor activity as described for 17α-hydroxypregnanolone-3-glutamate [83]. The possibility that 5β-pregnane-3α/β, α/β-diols may have their own pharmacology remains to be explored. Finally, the most potent PXR ligand is the unnatural steroid 5β-androstan-3α-ol, but the structural space around this lead has not been fully developed. Such ligands could be developed to induce CYP3A4 to prevent drug-induced liver injury.

Another frontier is to elucidate the pharmacology of 5β-reduced metabolites of anabolic steroids (17α-methyl-testosterone, fluoxymestrone and norethandrolone) and synthetic progestins (norethindrone and norgestrel) widely used in the oral contraceptive pill. 

## Figures and Tables

**Figure 1 ijms-25-08857-f001:**
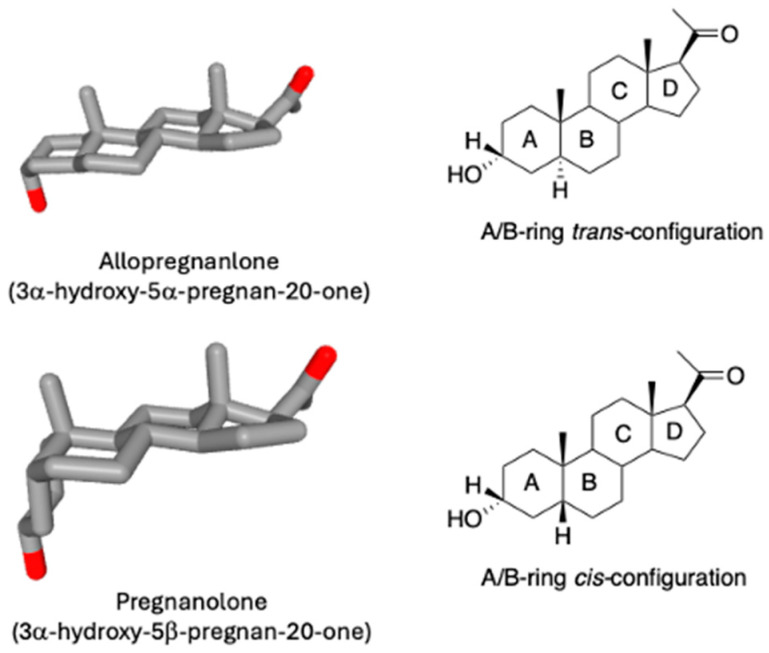
Bent steroid configuration seen in 5β-dihydrosteroids.

**Figure 2 ijms-25-08857-f002:**
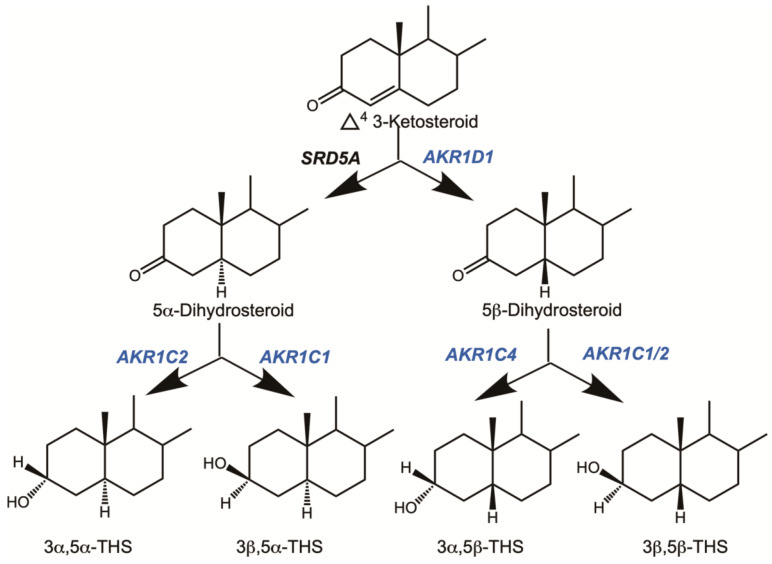
Metabolism of Δ^4^-3-ketosteroids to tetrahydrosteroids. The sequential role of aldo-keto reductases is illustrated. Reproduced with permission from Endocrine Society [5].

**Figure 3 ijms-25-08857-f003:**
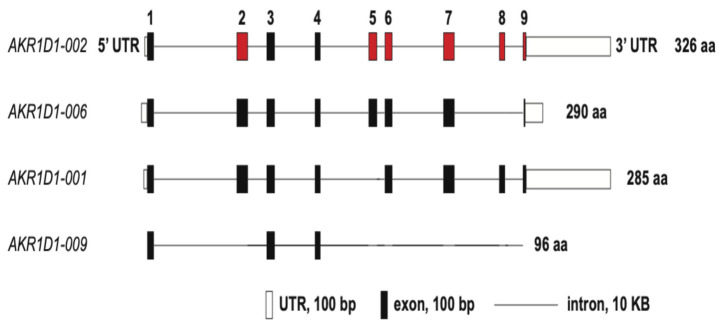
AKR1D1 splice variants. Reproduced with permission from *Steroids* [14].

**Figure 4 ijms-25-08857-f004:**
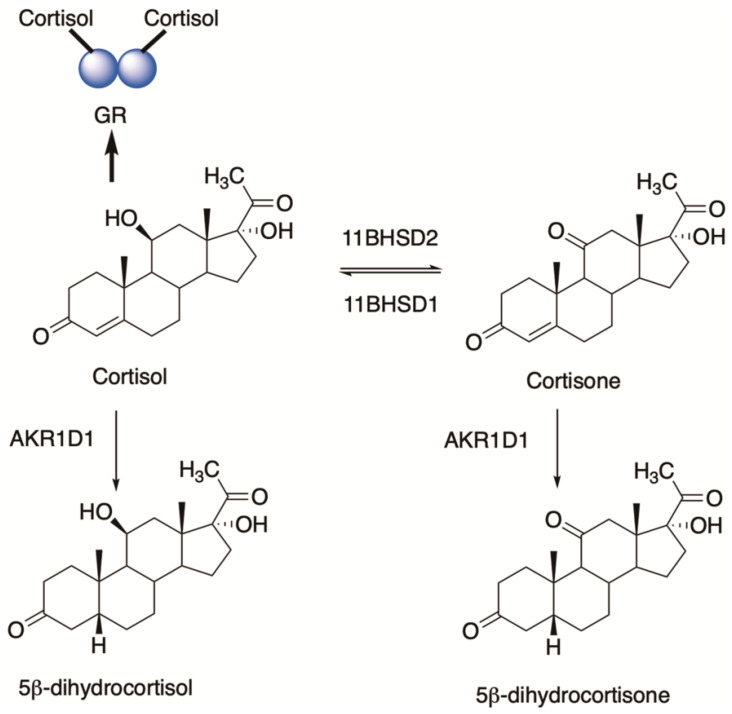
Control of ligand access to the glucocorticoid receptor in liver cells.

**Figure 5 ijms-25-08857-f005:**
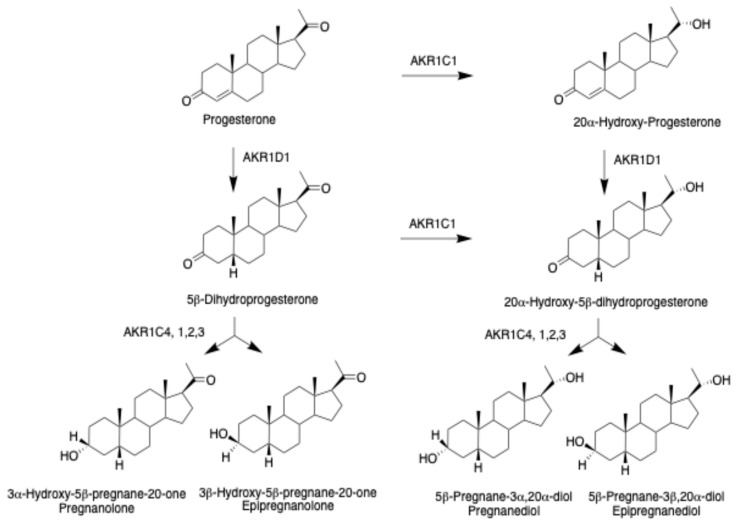
Biosynthesis of 5β-pregnanes from progesterone.

**Figure 6 ijms-25-08857-f006:**
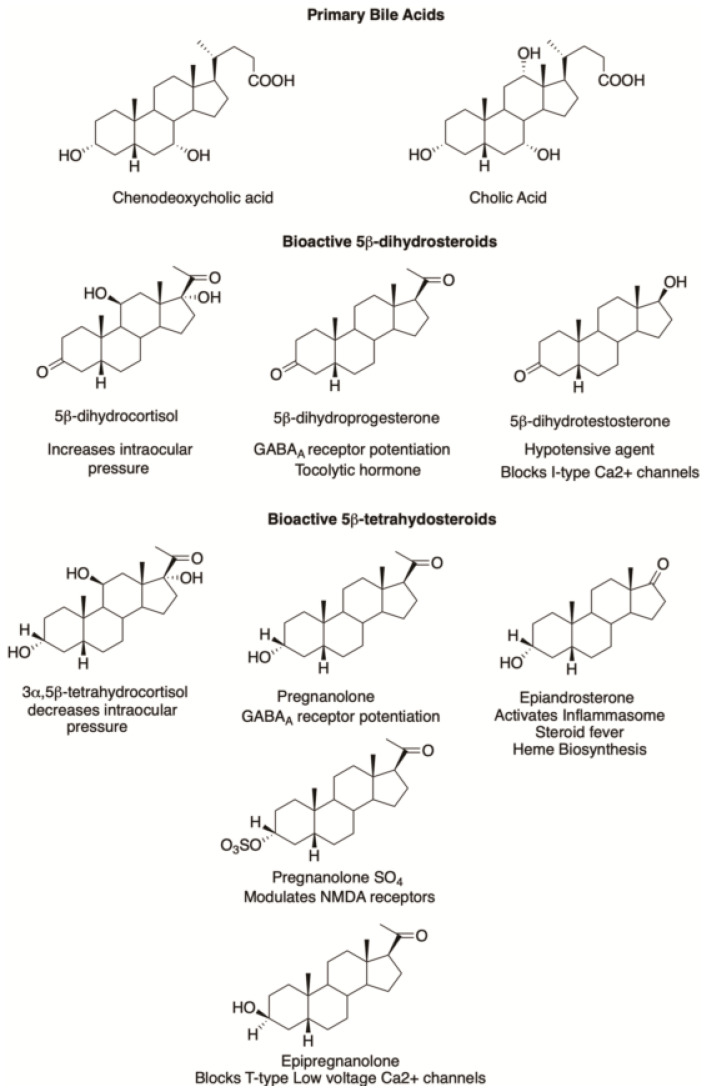
Bioactive 5β-dihydrosteroids.

**Figure 7 ijms-25-08857-f007:**
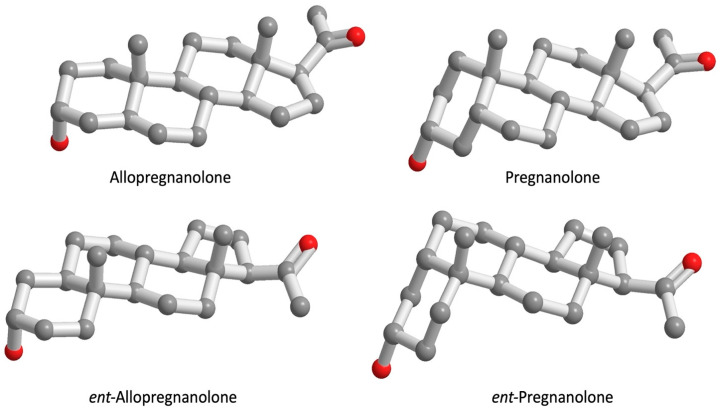
Allopregnanolone, *ent*-allopregnanolone and pregnanolone, *ent*-pregnanolone. The plane of the page is the mirror plane with allopregnanolone and pregnanolone behind the plane of the page and the *ent*-allopregnanolone and *ent*-pregnanolone in front of the plane of the page. Overlay of the respective enantiomer pairs would superimpose the A and C rings as well as the 18 and 19 methyl groups in each enantiomer pair.

**Figure 8 ijms-25-08857-f008:**
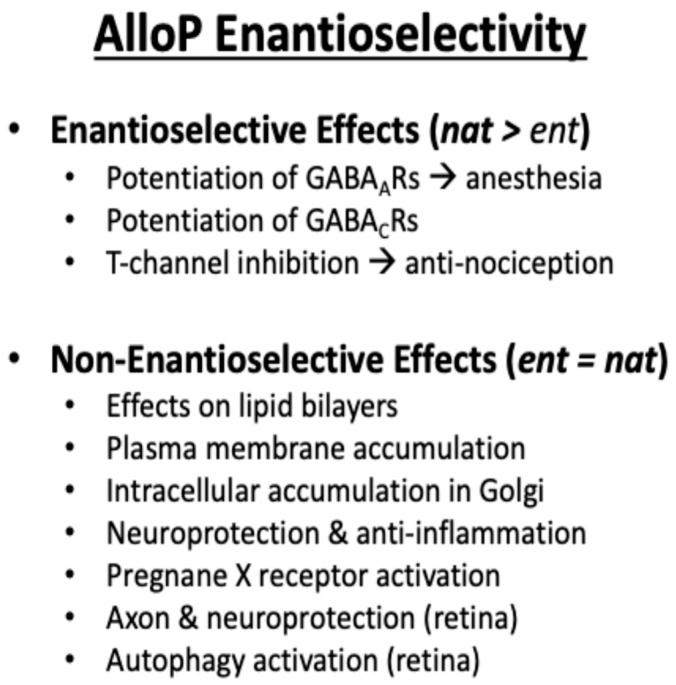
Properties of *ent*-steroids. The figure summarizes various effects where the enantiomers of AlloP (allopregnanolone) have been compared, including effects with enantioselectivity (*nat > ent*), and effects where the enantiomers are equivalent (*nat = ent*). Reproduced with permission from *Neuroscience Biohav. Res* [60].

**Table 1 ijms-25-08857-t001:** 5β-Steroid ligands that bind to the oxytocin receptor (OTR).

Inhibition of [^3^H] Oxytocin Binding to OTR by Progesterone Derivatives		
Steroid Compound	*K*_i_ (nM) *	
	Rat OTR	Human OTR
Progesterone	19 ± 3	None ^†^
R5020	59 ± 7	None ^†^
RU486	46 ± 5	None ^†^
5β-pregnane-3,20-dione	None ^†^	32 ± 5

* Membranes derived from CHO cells expressing either the rat or the human OTR were incubated with 2 nM [^3^H] oxytocin and the effect of steroids on oxytocin binding was expressed as the inhibition constant, Ki (mean ± SEM). ^†^ No inhibition was observable at concentrations up to 10 μM. Taken from [62].

**Table 2 ijms-25-08857-t002:** Effect of progesterone metabolites (10.0 μM) to increase [Ca^2+^]i (nM) in platelets.

Progesterone Metabolite	nM
4-Pregnene-3,20-dione (progesterone)	5 ± 4
5α-Pregnane-3α,21-diol-20-one (allo THDOC)	21 ± 2
5α-Pregnane-3,20-dione (5α-dihydroprogesterone)	26 ± 5
5α-Pregnan-3α-ol-20-one (allopregnanolone)	26 ± 4
4-Pregnen-20α-ol-3-one (20α-hydroxyprogesterone)	28 ± 4
5β-Pregnane-3α, 21-diol-20-one (THDOC)	44 ± 5
5α-Pregnan-3β-ol-20-one (isopregnanolone)	58 ± 19
5β-Pregnane-3,20-dione (5β-dihydroprogesterone)	68 ±14
5α-Pregnane-3α,20α-diol (allo-pregnanediol)	100 ± 10
5β-Pregnan-3β-ol-20-one (epipregnanolone)	107 ± 13
5β-Pregnan-3α-ol-20-one (pregnanolone)	142 ± 20
5α-Pregnane-3β,20α-diol (isopregnanediol)	215 ± 18
5β-Pregnane-3α,20α-diol (pregnanediol)	413 ± 116
Thrombin (0.01 U/mL)	417 ± 31

Data are expressed as increase in [Ca^2+^]i above basal [Ca^2+^]i in nM (mean ± S.E.M. from between three and six separate experiments). The effect of progesterone to increase [Ca^2+^]i was not significant (*p* > 0.05), all the other steroids in the table produced significant increases in [Ca^2+^] i (*p* < 0.05). All steroid concentrations were 10 μM. Taken from [66].

**Table 3 ijms-25-08857-t003:** 5β-Steroid ligands that activate the human PXR ^1^.

5β-Steroid	EC_50_ (μM)	% Efficacy/100
5β-pregnane-3,20-dione	2.6 ± 0.2	0.97
5β-pregnane-3α,20β-diol	3.81 ± 0.3	0.94
5β-androstan-3α-ol	1.41 ± 0.01	1.12

Lithocholic acid acetate	1.2 ± 0.2	0.54
Lithocholic acid	10.0 ± 0.1	0.15
7-Ketolithocholic acid	21.5 ± 1.4	0.58
12-Ketolithocholic acid	31.3 ± 5.8	0.86
Taurochenodeoxycholate	104 ± 0.8	0.5
5β-cholan-3α,7α,12α,24-tetrol	>100	0

^1^ Taken from [82].

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
