# Peer review of "5β-Dihydrosteroids: Formation and Properties"

_ijms, 2024, doi:10.3390/ijms25168857_

Round 1

Reviewer 1 Report

Comments and Suggestions for Authors

The review provides a comprehensive overview of the various 5beta-dihydrosteroids in the various categories of steroid hormones, their structures and the mode of action as ligands of nuclear and membrane-bound receptors.

The manuscript is well structured and well written. However, there are a few issues that have to be addressed. Please find my concerns and suggestions below.

Line 148/149: Shall it mean “…it is unknown whether the microbiota was able to compensate for the defect in primary bile acid synthesis or whether the effect was due to the expression of own steroid 5b-reductase.“ ?

Lines 236-238: When comparing with Figure 6 the sentences “Structure activity relationships demonstrate that the presence of the 3a-hydroxy group is essential for activity on the GABAA receptor [37]. 5b-Dihydroprogesterone or 3a,5b-tetrahydroprogesterone can modulate the GABAA receptor as well.” are confusing. I suggest to remove “or 3a,5b-tetrahydroprogesterone” from the second sentence.

Lines 240-242: The sentence “Structure activity relationships showed that 3a,5b-tetrahydroprogesterone (pregnanolone) had a less potentiating effect of allopregnanolone on flunitrazepam binding, Figure 6 [37]. “ is probably missing one word.

Line 275-278: in line 275 “M1” has probably to be added and I am missing a reference for the statements.

Lines 289-290: I would add “might” or “could” in front of “cross…”. Please add a reference for the statement.

Line 302: please cite here only Figure 7

Line 310: I would initiate the sentence “Effects on lipid…” with “However, effects…” or “On the other hand, effects…”

Line 312: please refer here to Figure 8

Line 330: I would phrase “However, others showed that …”

Lines 348-350: I have several difficulties with the sentence/statement “Structure -activity relationships demonstrate that both pregnanediol and isopregnanediol (5a-Pregnane-3b,20a-diol) were equally efficacious indicating that the A/B ring fusion was not important for this activity.”

-       Why is the full name of pregnanediol not also given – would make comparison for readers more easy

-       “A/B ring fusion” is a special expression/condition and would have to be explained to the reader or rephrased

-       When looking into the reference [64] I have the impression that the authors come to another conclusion than the authors of the manuscript here.

Line 366-373: Please indicate that the statements in this paragraph are taken from a review

Line 374: Please correct to “The 3a,5b-androgen etiocholanolone (3a-hydroxy-5b-androstane-17-one) also …”  

Line 387: I would change “of GABAA receptors” to “of the GABAA receptor”

Line 390: I would rephrase chapter title to „Farnesoid X Receptor (FXR) ligands“

Line 391: I would remove here the PXR and write “5b-Dihydrosteroids can act as ligands for the nuclear receptor FXR”

Line 401: I would change “preferred” to “better inducers”

Line 404: Please rephrase chapter title to “Pregnane X Receptor (PXR) ligands”

Then I would introduce the chapter with “5b-Dihydrosteroids can also act as ligands for the nuclear receptor PXR”

Line 410: I would add “for example” behind “such as”

Line 411: remove “see”

Lines 411-412: When looking at Table 3 and comparing with a statement in the discussion the sentence “5b-Androstan-3a-ol is almost….” should read “5b-Androstan-3a-ol is even more potent efficacious as 5β-pregnane-3,20-dione but it is not an endogenous steroid.”

Line 415: I would begin the sentence “There are distinct…” with “However, there are …”

Line 417: please remove “but”

Line 434-435: Shall it read “Evidence for the de novo synthesis of D4-3-ketosteroids to form NAS in the CNS has been controversial.”?

Table 1

In the legend, 10 mM has to be corrected to 10 µM

Figure 6

below the structures it is written GABAa, please correct to GABAA.

Figure 8

In the legend, AlloP should be fully named

Author Response

Reviewer #1.

The review provides a comprehensive overview of the various 5beta-dihydrosteroids in the various categories of steroid hormones, their structures and the mode of action as ligands of nuclear and membrane-bound receptors.

The manuscript is well structured and well written. However, there are a few issues that have to be addressed. Please find my concerns and suggestions below.

Comment: We thank the reviewer for these positive comments.

Line 148/149: Shall it mean “…it is unknown whether the microbiota was able to compensate for the defect in primary bile acid synthesis or whether the effect was due to the expression of own steroid 5b-reductase.“ ?

Corrected.

Lines 236-238: When comparing with Figure 6 the sentences “Structure activity relationships demonstrate that the presence of the 3a-hydroxy group is essential for activity on the GABAA receptor [37]. 5b-Dihydroprogesterone or 3a,5b-tetrahydroprogesterone can modulate the GABAA receptor as well.” are confusing. I suggest to remove “or 3a,5b-tetrahydroprogesterone” from the second sentence.

Corrected. Now Line 255.

Lines 240-242: The sentence “Structure activity relationships showed that 3a,5b-tetrahydroprogesterone (pregnanolone) had a less potentiating effect of allopregnanolone on flunitrazepam binding, Figure 6 [37]. “ is probably missing one word.

Corrected. Now line 260.

Line 275-278: in line 275 “M1” has probably to be added and I am missing a reference for the statements.

Corrected. Now line 293.

Lines 289-290: I would add “might” or “could” in front of “cross…”. Please add a reference for the statement.

Corrected. Now line 314.

Line 302: please cite here only Figure 7

Corrected. Now line 327.

Line 310: I would initiate the sentence “Effects on lipid…” with “However, effects…” or “On the other hand, effects…”

Corrected. Now line 335.

Line 312: please refer here to Figure 8

Corrected. Now line 337.

Line 330: I would phrase “However, others showed that …”

Corrected. Now line 351.

Lines 348-350: I have several difficulties with the sentence/statement “Structure -activity relationships demonstrate that both pregnanediol and isopregnanediol (5a-Pregnane-3b,20a-diol) were equally efficacious indicating that the A/B ring fusion was not important for this activity.”

-       Why is the full name of pregnanediol not also given – would make comparison for readers more easy

Corrected. Now line 382.

-       “A/B ring fusion” is a special expression/condition and would have to be explained to the reader or rephrased

Rephrased. Now line 383.

-       When looking into the reference [64] I have the impression that the authors come to another conclusion than the authors of the manuscript here.

This is stimulate Ca2+ influx which leads to platelet activation. Line 380.

Line 366-373: Please indicate that the statements in this paragraph are taken from a review

References Cited: Lines 400-411.

Line 374: Please correct to “The 3a,5b-androgen etiocholanolone (3a-hydroxy-5b-androstane-17-one) also …”

Corrected. Now Line 415.  

Line 387: I would change “of GABAA receptors” to “of the GABAA receptor”

Corrected. Now line 428.

Line 390: I would rephrase chapter title to „Farnesoid X Receptor (FXR) ligands“

Corrected. Now line 431.

Line 391: I would remove here the PXR and write “5b-Dihydrosteroids can act as ligands for the nuclear receptor FXR”

Corrected. Now line 432.

Line 401: I would change “preferred” to “better inducers”

Corrected. Now line 454.

Line 404: Please rephrase chapter title to “Pregnane X Receptor (PXR) ligands”

Then I would introduce the chapter with “5b-Dihydrosteroids can also act as ligands for the nuclear receptor PXR”

Corrected. Now line 457.

Line 410: I would add “for example” behind “such as”

Corrected. Now line 464.

Line 411: remove “see”

Corrected. Now line 464.

Lines 411-412: When looking at Table 3 and comparing with a statement in the discussion the sentence “5b-Androstan-3a-ol is almost….” should read “5b-Androstan-3a-ol is even more potent efficacious as 5β-pregnane-3,20-dione but it is not an endogenous steroid.”

Corrected. Now line 465.

Line 415: I would begin the sentence “There are distinct…” with “However, there are …”

Corrected. Now line 469.

Line 417: please remove “but”

Corrected. Now line 471.

Line 434-435: Shall it read “Evidence for the de novo synthesis of D4-3-ketosteroids to form NAS in the CNS has been controversial.”?

Corrected. Lines 494-495.

Table 1

In the legend, 10 mM has to be corrected to 10 µM

Corrected.

Figure 6

below the structures it is written GABAa, please correct to GABAA.

Corrected.

Figure 8

In the legend, AlloP should be fully named

Corrected.

Reviewer #2

The manuscript submitted by Penning and Covey for publication in International Journal of Molecular Sciences aims at studying synthesis and properties of 5β-dihydrosteroids. These steroids have not been extensively studied, and there is a notable lack of information about them, making this review an important contribution to the field.

This manuscript describes the result of considerable effort. This is an interesting study of general significance that will make a useful addition to the literature. Overall the manuscript is well-written, easy to follow and logically structured. I found no major concerns.

 Comment: We thank the reviewer for these very positive comments.

I have only several additional comments and I suggest only a minor revision before publication.

 See responses below

  1. Figures 3 and 8 should be in better resolution.

Higher resolution Figures have been provided. We thank the reviewer for this suggestion.

  1. The authors should consider dividing section 7 into subsection.

We now provide subsections to section 7. We thank the reviewer for this suggestion.

  1. In Table 1. information about what ± value represents is missing.

Defined.

  1. Line 370: “.“ should be deleted.

Deleted.

  1. The authors should consider adding more information about FXR in Chapter 9.

Section has been expanded. See lines 436-441.

  1. The authors should include information in the manuscript regarding the significance of identifying modulators of 5β-reductase (AKR1D1) activity.

Section expanded. See lines 182-197.

Reviewer 2 Report

Comments and Suggestions for Authors

5β-Dihydrosteroids: Formation and Properties

The manuscript submitted by Penning and Covey for publication in International Journal of Molecular Sciences aims at studying synthesis and properties of 5β-dihydrosteroids. These steroids have not been extensively studied, and there is a notable lack of information about them, making this review an important contribution to the field.

This manuscript describes the result of considerable effort. This is an interesting study of general significance that will make a useful addition to the literature. Overall the manuscript is well-written, easy to follow and logically structured. I found no major concerns.

I have only several additional comments and I suggest only a minor revision before publication.

1. Figures 3 and 8 should be in better resolution.

2. The authors should consider dividing section 7 into subsection.

3. In Table 1. information about what ± value represents is missing.

4. Line 370: “.“ should be deleted.

5. The authors should consider adding more information about FXR in Chapter 9.

6. The authors should include information in the manuscript regarding the significance of identifying modulators of 5β-reductase (AKR1D1) activity.

Thank you for the opportunity to read this interesting study

Author Response

Reviewer #2

The manuscript submitted by Penning and Covey for publication in International Journal of Molecular Sciences aims at studying synthesis and properties of 5β-dihydrosteroids. These steroids have not been extensively studied, and there is a notable lack of information about them, making this review an important contribution to the field.

This manuscript describes the result of considerable effort. This is an interesting study of general significance that will make a useful addition to the literature. Overall the manuscript is well-written, easy to follow and logically structured. I found no major concerns.

 Comment: We thank the reviewer for these very positive comments.

I have only several additional comments and I suggest only a minor revision before publication.

 See responses below

  1. Figures 3 and 8 should be in better resolution.

Higher resolution Figures have been provided. We thank the reviewer for this suggestion.

  1. The authors should consider dividing section 7 into subsection.

We now provide subsections to section 7. We thank the reviewer for this suggestion.

  1. In Table 1. information about what ± value represents is missing.

Defined.

  1. Line 370: “.“ should be deleted.

Deleted.

  1. The authors should consider adding more information about FXR in Chapter 9.

Section has been expanded. See lines 436-441.

  1. The authors should include information in the manuscript regarding the significance of identifying modulators of 5β-reductase (AKR1D1) activity.

Section expanded. See lines 182-197.

Round 2

Reviewer 1 Report

Comments and Suggestions for Authors

All my earlier concerns have been addressed in the current revised manuscript. There are only some minor formal issues left:

Lines 148, 384:  spaces have to be removed

Lines 400, 409, 411: reference citation in the text must be unified: dot or comma behind the references

Figure 6: “GABAa” has to be exchanged by “GABAA”

Reviewer 2 Report

Comments and Suggestions for Authors

The authors have carefully answered to my questions and I believe that the paper is now ready for publication. 
